**SciPost Physics Community Reports** Submission

# $gg \to ZH$ : updated predictions at NLO QCD

Benjamin Campillo Aveleira[1†], Long Chen[2∨], Joshua Davies[3⊢], Giuseppe Degrassi[4††], Pier Paolo Giardino[5‖], Ramona Gröber[6⋆], Gudrun Heinrich[1‡], Stephen Jones[7¶], Matthias Kerner[1∘], Johannes Schlenk[8▷], Matthias Steinhauser[9◁] and Marco Vitti[9,10∧]

**1** Institute for Theoretical Physics, KIT, 76131 Karlsruhe, Germany
**2** School of Physics, Shandong University, Jinan, 250100 Shandong, China
**3** Department of Mathematical Sciences, University of Liverpool, Liverpool, UK
**4** Dipartimento di Matematica e Fisica, Università di Roma Tre, and INFN, Sezione di Roma Tre, Rome, Italy
**5** Departamento de Física Teórica and Instituto de Física Teórica UAM/CSIC, Universidad Autónoma de Madrid, Cantoblanco, 28049, Madrid, Spain
**6** Dipartimento di Fisica e Astronomia 'G.Galilei', Università di Padova and INFN, Sezione di Padova, Padua, Italy
**7** Institute of Particle Physics Phenomenology, Durham University, Durham, UK
**8** Department of Astrophysics, University of Zurich, 8057 Zurich, Switzerland
**9** Institute for Theoretical Particle Physics, KIT, 76131 Karlsruhe, Germany
**10** Institute for Astroparticle Physics, KIT, 76344 Karlsruhe, Germany

† benjamin.campillo@kit.edu , ∨ longchen@sdu.edu.cn , ⊢ J.O.Davies@liverpool.ac.uk ,
†† giuseppe.degrassi@uniroma3.it , ‖ pier.giardino@uam.es , ⋆ ramona.groeber@unipd.it ,
‡ gudrun.heinrich@kit.edu , ¶ stephen.jones@durham.ac.uk , ▷ johannes.schlenk@psi.ch ,
∘ matthias.kerner@kit.edu , ◁ matthias.steinhauser@kit.edu , ∧ marco.vitti@kit.edu

## Abstract

**We present state-of-the-art predictions for the inclusive cross section of gluon-initiated *ZH* production, following the recommendations of the LHC Higgs Working Group. In particular, we include NLO QCD corrections, where the virtual corrections are obtained from the combination of a forward expansion and a high-energy expansion, and the real corrections are exact. The expanded results for the virtual corrections are compared in detail to full numerical results. The updated predictions show a reduction of the scale uncertainties to the level of 15%, and they include an estimate of the top-mass-scheme uncertainty.**

# 1 Introduction

One of the main goals of the Large Hadron Collider (LHC) physics program is a precise under-standing of the Higgs boson. Apart from the ongoing experimental activities to pin down the properties of the Higgs boson [1, 2], this requires also a profound theoretical understanding of Higgs production and decays within the Standard Model (SM) and beyond. Among the rel-evant Higgs production modes, the associated $VH$ production of a Higgs boson with a vector boson, $V = W^{\pm}$ or $Z$, provides the third largest cross section for Higgs boson production at the LHC. In addition, it is the most convenient process to measure the Higgs coupling to bottom quarks [3–5] and possibly to charm quarks [6, 7], which are difficult to access both in Higgs production via gluon fusion and vector boson fusion due to large backgrounds.

  With the ever-increasing precision of the LHC measurements, higher-order calculations are essential to obtain agreement between the measurements and theory predictions. Considering the quark-initiated channel for $VH$ production, $q\bar{q} \to VH$, the inclusive cross section is known at next-to-next-to-next-to-leading order (N$^3$LO) in the strong coupling, the topologies being of Drell-Yan type [8], superseding the NNLO QCD results of Refs. [9, 10] that are available in the public code VH@NNLO [11, 12]. Fully differential results are available at NNLO [13, 14], including also the decay of the Higgs boson to bottom quarks [15, 16] and anomalous couplings [17]. Electroweak corrections have been computed in Refs. [18,19] and are available in the code HAWK [20].

  Currently, the scale uncertainties of $WH$ production quoted in the Yellow Report 4 (YR4) [21] are at the 1% level, while for the $ZH$ process they are roughly three times larger. The reason is that at NNLO QCD, $ZH$ production receives contributions from a gluon-initiated sub-process via one-loop diagrams [22,23], which is affected by $\mathcal{O}(25)\%$ scale uncertainties. The NNLO suppression of the $gg$-initiated channel is compensated by the large gluon luminosity, which leads to a significant impact on the overall uncertainty compared to $WH$ production, where such contributions are forbidden by charge conservation. The $gg$-initiated channel gains relative importance with respect to the $q\bar{q}$ channel in the boosted regime [24]. In or-der to reduce the scale uncertainties in $ZH$ production, it is important to have predictions for $gg \to ZH$ at NLO accuracy in QCD.

  The calculation of the NLO QCD corrections to $gg \to ZH$ requires the evaluation of two-loop multi-scale integrals with massive internal lines, because the dominant contribution to the amplitude involves loops of top quarks. The NLO corrections were first computed in the infinite-top-mass limit [25] and augmented with an expansion in small external momentum with respect to the top-quark mass [26,27], where Ref. [27] provides also results in the high-

energy regime. However, both expansions do not cover completely the relevant phase space of the process, which motivates the computation retaining the full dependence on the top-quark mass. The two-loop virtual corrections were computed numerically with full top-quark-mass dependence in Ref. [28]. In Ref. [29] they were calculated numerically in an expansion in small external masses and in Ref. [30] using an analytic approach via an expansion in small transverse momentum, $p_T$, based on Ref. [31]. The expansion in Ref. [30] covers the region of the phase space complementary to the expansion in the high-energy limit of Ref. [27]. Total and differential cross sections at NLO QCD were provided in Refs. [32, 33], soft gluon resummation at NLL has been performed in Refs. [34, 35], where Ref. [35] also contains subleading contributions and differential results.

The work presented here is based on Refs. [32, 33]. We performed a detailed comparison of the corresponding results and provide recommendations in the context of the Report 5 of the LHC Higgs Working Group. The rest of the report is structured as follows: in Section 2 we discuss the different approaches that entered the calculation of the NLO QCD corrections and compare them. In Section 3 we provide fixed-order predictions for the inclusive cross section at NLO QCD and discuss the theoretical uncertainties, before we conclude in Section 4.

## 2    Calculation of NLO QCD corrections

Based on a combination of the two approaches [32] and [33], described briefly in the following, we have implemented the NLO QCD corrections into a Monte Carlo code ggHZ using the POWHEG-Box-V2 framework [36, 37]. The details of the implementation will be discussed in a separate publication [38]. In this contribution, we mainly report the results for the inclusive cross sections that have been obtained using the ggHZ code.

### 2.1   Virtual corrections

The two calculations [28, 32] and [33] follow two very different approaches with regard to the computation of the virtual corrections.

In Ref. [28], the two-loop virtual amplitudes are calculated numerically with full top-quark-mass dependence. The amplitude has been generated using QGraf [39] and FORM [40, 41] and reduced to master integrals based on five planar and three non-planar integral families using Kira-2.0 [42] and FireFly [43] for the reduction and Reduze [44] and LiteRed [45] for the rotation to a quasi-finite basis [46]. To ease the IBP reduction, the ratios $m_H^2/m_t^2 = 12/23$ and $m_Z^2/m_t^2 = 23/83$ have been used. The master integrals have been calculated numerically with the program pySecDec [47–49]. The independent helicity amplitudes that contribute to the final result have been calculated individually.

Instead, the computation of the virtual corrections in Ref. [33] is based on an expansion in small transverse momentum [30] for the box-type contributions, which is valid for values of the partonic Mandelstam variables such that $|\hat{t}| \leq 4m_t^2$ or $|\hat{u}| \leq 4m_t^2$. The expansion reduces the number of scales in the multi-scale integrals to one, which allows to express the master integrals[1] analytically in terms of generalised harmonic polylogarithms (GHPLs). Only two elliptic integrals could not be expressed in terms of GHPLs and are obtained as series expansion around singular and intermediate points [51]. For what regards the virtual corrections to the triangle-type diagrams, they can be obtained (as observed in Ref. [25]) from the computation of pseudo-scalar-Higgs production in gluon fusion of Refs. [52, 53], and we use the implementation of Ref. [54]. Finally, virtual corrections coming from one-particle-reducible

---

[1]The master integrals are exactly the same as for the NLO QCD corrections of the top-quark loops to $gg \rightarrow HH$ [31] and $gg \rightarrow ZZ$ [50].

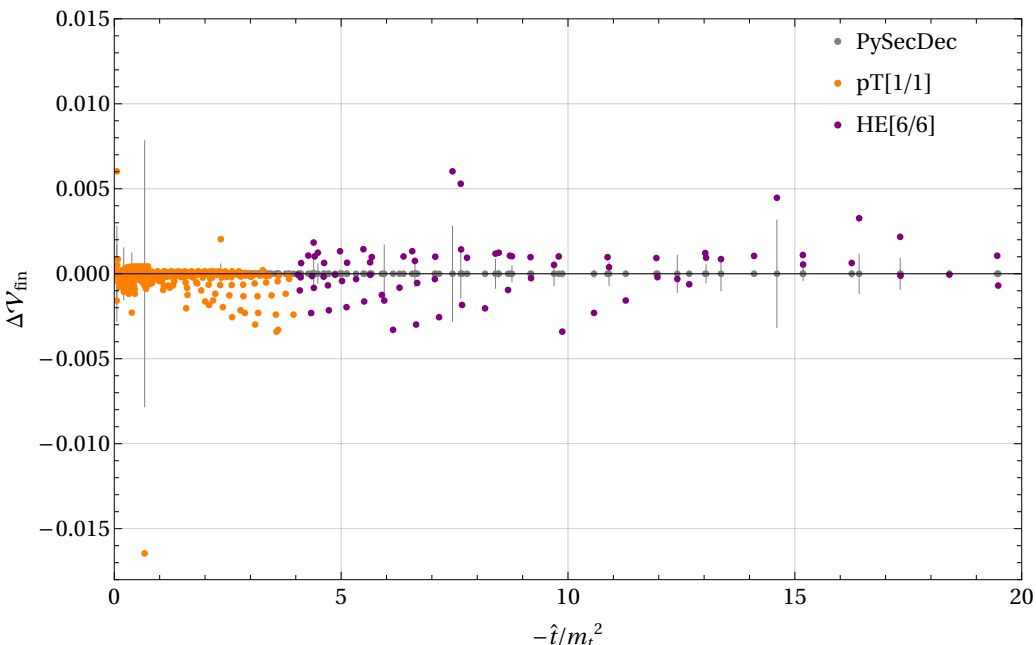

Figure 1: Comparison of the results for $\mathcal{V}_{\text{fin}}$ of Ref. [33] and Ref. [28]. The orange (purple) points denote the normalized difference defined in Eq. (2), where the $p_T$ (high-energy) expansion is used in $\mathcal{V}_{\text{fin}}^{\text{Ref. [33]}}$. The grey error bars indicate the relative uncertainty associated to the points from $\mathcal{V}_{\text{fin}}^{\text{Ref. [28]}}$. The figure is based on a comparison of 460 different phase space points $(\hat{s}, \hat{t})$. The range of $\hat{t}$ values corresponds to $p_T < 670$ GeV.

double-triangle diagrams are calculated in exact form.

Since the expansion in small $p_T$ is not valid for the whole phase space, it has to be augmented with an expansion in the high-energy regime as provided in Ref. [27], which covers the complementary phase space. The two expansions are combined after being improved via Padé approximants, as detailed in Ref. [55]. We use a [1/1] Padé for the $p_T$ expansion and a [6/6] Padé for the high-energy expansion, which provide stable results in the region $|\hat{t}| \simeq 4m_t^2$ (or $|\hat{u}| \simeq 4m_t^2$).

In order to produce the numerical results presented here, we use the combination of the $p_T$ expansion and the high-energy expansion presented in Ref. [33]. While the numerical results are more accurate overall, the reason to choose the analytic approach is that its flexibility allows us to assess the uncertainty associated with the choice of the top mass renormalisation scheme. The UV-finite, IR-subtracted virtual corrections are defined as

$$\mathcal{V}_{\text{fin}} = \frac{G_\mu^2 m_Z^2}{16} \left( \frac{\alpha_s(\mu_R)}{\pi} \right)^2 \left\{ \sum_i \left| \mathcal{A}_i^{(0)} \right|^2 C_A \left( \pi^2 - \log^2 \left( \frac{\mu_R^2}{M_{ZH}^2} \right) \right) + 2 \sum_i \text{Re} \left( \mathcal{A}_i^{(0)} \mathcal{A}_i^{(1)*} \right) \right\}, \quad (1)$$

where $\mathcal{A}_i^{(0)}$ and $\mathcal{A}_i^{(1)}$ are LO and NLO form factors obtained from a decomposition of the amplitude in terms of projectors. The $\mathcal{A}_i^{(0)}$ are computed in full analytic form, while the $\mathcal{A}_i^{(1)}$ are obtained from the combination of expansions described above.

The results in the approach of Ref. [33] have been checked against the results of Ref. [32], which have been obtained from a combination of the numerical evaluation of Ref. [28] and the high-energy expansion. The comparison of 460 different phase-space points is summarised in

Fig. 1, where the quantity

$$\Delta \mathcal{V}_{\text{fin}} = \frac{\alpha_{\text{s}}}{4\pi} \frac{\left(\mathcal{V}_{\text{fin}}^{\text{Ref. [28]}} - \mathcal{V}_{\text{fin}}^{\text{Ref. [33]}}\right)}{\mathcal{B}} \tag{2}$$

represents the difference between the two calculations of $\mathcal{V}_{\text{fin}}$ normalized to the Born result, denoted as $\mathcal{B}$, which is insensitive to the choice of the infra-red subtraction scheme. We fix $\alpha_{\text{s}} = 0.118$ in Eq. (2). We observe that differences between the two approaches mostly do not exceed a few per mille for $|\hat{t}| \lesssim 4m_t^2$, while slightly larger differences can be found when increasing $|\hat{t}|$: this is attributed to the fact that $\mathcal{V}_{\text{fin}}^{\text{Ref. [33]}}$ includes only terms up to $\mathcal{O}(m_H^2, m_Z^2)$ in the high-energy expansion. As already shown in Ref. [32], higher order terms in $m_H^2$ and $m_Z^2$ in this expansion improve the agreement by more than one order of magnitude. In total, we can conclude that using the analytic results of Refs. [27, 30, 55] as base for our recommendation will lead to differences within 1% with respect to the virtual corrections with exact top-mass dependence of Ref. [28]. This difference is negligible with respect to other theoretical uncertainties that will be discussed in Section 3.

We note that a deeper expansion around the forward limit has recently been obtained [56, 57]. Following the approach of Ref. [58], the combination with the high-energy expansion including quartic terms in $m_H$ and $m_Z$ and more than 100 expansion terms in $m_t$ leads to an agreement with the numerical results of Ref. [32] that is better than per mille. A publication is in preparation [57]. It is also planned to implement the result in the library ggxy [59].

## 2.2 Real emission contributions

The real corrections in Refs. [32, 33] were obtained using different tools: Ref. [32] uses GoSam [60, 61], while Ref. [33] uses Recola2 [62, 63]. The results were compared for a large number of phase-space points and we found an agreement below the per mille level for the contribution of each partonic channel[2]. After this validation, we regenerated the real corrections, with the newest version of GoSam [64] used to provide the numerical values for the cross sections presented here. The new results were compared in detail to the original work of Ref. [32].

### 2.2.1 $Z$-boson radiation

As pointed out in Ref. [33], as an additional contribution to the real corrections, there are diagrams where a $Z$ boson is radiated from an open quark line. Representative $Z$-radiation Feynman diagrams are shown in Fig. 2 for the $qg$ and $q\bar{q}$ channels. These diagrams were computed using GoSam and cross-checked at the level of the total cross section with Ref. [33]. The $Z$-radiation diagrams have not been included in the calculation of Ref. [32] because they were considered as Drell-Yan-type contributions. However, so far the $\mathcal{O}(\alpha_s^3)$ contributions stemming from the square of the $Z$-radiation diagrams have not been included into the calculation of the Drell-Yan-like $ZH$ production. For this reason we include them into the $gg \to ZH$ component. We also note that the $\mathcal{O}(\alpha_s^2)$ interference between these diagrams and the Drell-Yan-type real radiation diagrams is not included in our calculation, in order to avoid a double counting in the combination of our results with those of Ref. [11].

## 2.3 Assessment of top-mass scheme uncertainty

The value of the top-quark mass can be chosen freely in the implementation of the NLO corrections of Ref. [33], which allows the uncertainty stemming from the choice of the renormalisation scheme for the top-quark mass to be estimated. We follow the approach of Ref. [65]

---

[2]We thank Xiaoran Zhao for assisting us with this cross check.

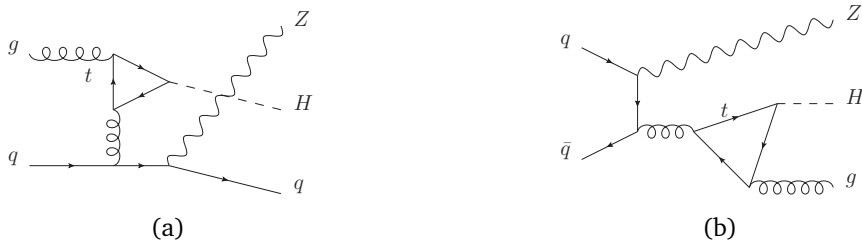

(a)                 (b)

Figure 2: Representative $Z$-radiation Feynman diagrams for the real-emission contributions in the $qg$ and $q\bar{q}$ channels.

employed for Higgs boson pair production, which takes the result in the on-shell scheme (OS) as central value and defines the uncertainty as the envelope of the predictions where the top-quark mass is evaluated in the $\overline{\text{MS}}$ scheme at scales $\mu_t = \{M_{ZH}, M_{ZH}/2, M_{ZH}/4\}$. We note that this is a conservative approach, which is likely to overestimate the uncertainty. In the context of Higgs boson pair production, it was recently shown that at leading power the leading mass logarithms in the high-energy limit can be resummed, decreasing the difference between the $\overline{\text{MS}}$ and the OS result at very large invariant masses [66]. For the $gg \to ZH$ process, the leading logarithms appear already at next-to-leading power and the resummation is therefore less straightforward [32].

The different predictions for the top-quark mass have been computed with CRunDec [67, 68], which incorporates the computation of $\alpha_s$ with six active flavours from the knowledge of $\alpha_s$ with five active flavours from LHAPDF [69], the computation of the top-quark mass at a given scale and the running of $\alpha_s$ at five loops. The conversion between the on-shell mass $m_t$ and the $\overline{\text{MS}}$-mass $\overline{m}_t(\mu_t)$ is computed at four-loop order via the relation

$$\frac{\overline{m}_t(\mu_t = m_t)}{m_t} = 1 - 0.4244\,\alpha_s(m_t) - 0.9246\,\alpha_s^2(m_t) - 2.593\,\alpha_s^3(m_t) - (8.949 \pm 0.018)\,\alpha_s^4(m_t), \tag{3}$$

where the computation of the coefficients can be found in Ref. [70].

## 3 Predictions

### 3.1 Setup and numerical input

We follow the instructions of the LHC Higgs Working Group collected in https://twiki.cern.ch/twiki/bin/view/LHCPhysics/LHCHWG136TeVxsec. In particular, for the parameters relevant to our calculation, we use

$$G_\mu = 1.16637 \times 10^{-5}\ \text{GeV}^{-2}, \quad \alpha_s(m_Z) = 0.1180, \tag{4}$$
$$m_t = 172.5\ \text{GeV}, \quad m_Z = 91.1876\ \text{GeV},$$

where the value for the top-quark mass is taken as the on-shell value.
We perform a scan over the values of the hadronic centre-of-mass energy $\sqrt{S} \in \{13, 13.6, 14\}$ TeV and over different values of the Higgs mass. We use the PDF4LHC21_40_pdfas set [69] for the parton distribution functions. The central factorisation and renormalisation scales are taken as half of the invariant mass of the $ZH$ system,

$$\mu_F = \mu_R = \frac{M_{ZH}}{2}. \tag{5}$$

The scale uncertainties are obtained from a 7-point variation in the range $\mu_R/\mu_F \in [1/2, 2]$. We note that the choices for the central scale and for the range of variation are different than

| $\sqrt{S}$ [TeV] | $m_H$ [GeV] | $\sigma_{\text{LO}}$ [fb] | $\sigma_{\text{NLO}}$ [fb] |
|---|---|---|---|
| 13 | 124.6 | $64.3^{+27+0\%}_{-20-23\%}$ | $119.0^{+17+0\%}_{-14-13\%}$ |
| 13 | 125.0 | $64.0^{+27+0\%}_{-20-23\%}$ | $118.4^{+17+0\%}_{-14-13\%}$ |
| 13 | 125.09 | $63.9^{+27+0\%}_{-20-23\%}$ | $118.3^{+17+0\%}_{-14-13\%}$ |
| 13 | 125.38 | $63.7^{+27+0\%}_{-20-24\%}$ | $117.9^{+17+0\%}_{-14-13\%}$ |
| 13 | 125.6 | $63.6^{+27+0\%}_{-20-24\%}$ | $117.6^{+17+0\%}_{-14-13\%}$ |
| 13 | 126.0 | $63.3^{+27+0\%}_{-20-24\%}$ | $117.1^{+16+0\%}_{-14-13\%}$ |
| 13.6 | 124.6 | $70.9^{+26+0\%}_{-20-24\%}$ | $131.1^{+16+0\%}_{-14-13\%}$ |
| 13.6 | 125.0 | $70.6^{+26+0\%}_{-20-24\%}$ | $130.5^{+16+0\%}_{-14-13\%}$ |
| 13.6 | 125.09 | $70.5^{+26+0\%}_{-20-24\%}$ | $130.4^{+16+0\%}_{-14-13\%}$ |
| 13.6 | 125.38 | $70.3^{+26+0\%}_{-20-24\%}$ | $130.0^{+16+0\%}_{-14-13\%}$ |
| 13.6 | 125.6 | $70.1^{+26+0\%}_{-20-24\%}$ | $129.7^{+16+0\%}_{-14-13\%}$ |
| 13.6 | 126.0 | $69.8^{+26+0\%}_{-20-24\%}$ | $129.1^{+16+0\%}_{-14-13\%}$ |
| 14 | 124.6 | $75.5^{+26+0\%}_{-20-24\%}$ | $139.5^{+16+0\%}_{-14-14\%}$ |
| 14 | 125.0 | $75.2^{+26+0\%}_{-20-24\%}$ | $138.9^{+16+0\%}_{-14-13\%}$ |
| 14 | 125.09 | $75.1^{+26+0\%}_{-20-24\%}$ | $138.8^{+16+0\%}_{-14-13\%}$ |
| 14 | 125.38 | $74.9^{+26+0\%}_{-20-24\%}$ | $138.3^{+16+0\%}_{-14-13\%}$ |
| 14 | 125.6 | $74.7^{+26+0\%}_{-20-24\%}$ | $138.0^{+16+0\%}_{-14-13\%}$ |
| 14 | 126.0 | $74.4^{+26+0\%}_{-20-24\%}$ | $137.3^{+16+0\%}_{-14-14\%}$ |

Table 1: Inclusive cross-section for $gg \rightarrow ZH$: the format of the results is $\sigma \pm \Delta_{\text{scale}}[\%] \pm \Delta_{m_t}[\%]$.

the ones recommended for $VH$ production, namely $M_{ZH}$ as central scale and a variation by a factor 3. Our preference for $M_{ZH}/2$ as central scale is justified by the observation that this seems to cover the next-to-leading-log resummed soft gluon contributions [34]. The factor 2 for the variation of $\mu_F$ and $\mu_R$ is chosen in analogy with the recommendations for $gg \rightarrow H$ and $gg \rightarrow HH$, for which higher-order corrections have a similar impact as in $gg \rightarrow ZH$. Specifically, we assume that the scale uncertainties at NLO are reliable in accounting for the missing NNLO corrections, whereas the same is not true at LO with the NLO terms.

## 3.2 Numerical Results

The results for the inclusive cross section are presented in Tab. 1. We observe that the NLO cross section is about a factor 1.85 larger than the LO prediction. This relative factor is basically insensitive to the Higgs mass and to the hadronic center-of-mass energy. The size of the NLO corrections is compatible with that of similar gg-initiated processes such as $gg \rightarrow H$ and $gg \rightarrow HH$. The absolute value of the cross section increases by roughly 16% when going from 13 TeV to 14 TeV. Finally, a variation of the value of the Higgs mass in the recommended range leads to differences in the cross sections that are below 2%, both at LO and at NLO. We recall that in the YR4 [21] approximated NLO predictions for $gg \rightarrow ZH$ are obtained via a rescaling of the LO results by the $K$-factor in the infinite-top-mass limit from Ref. [25]: for the values of $\sqrt{S}$ presented here, the central value of the rescaled $m_t \rightarrow \infty$ result is well within the uncertainties of our updated prediction, showing that this approach is very effective at the inclusive level.

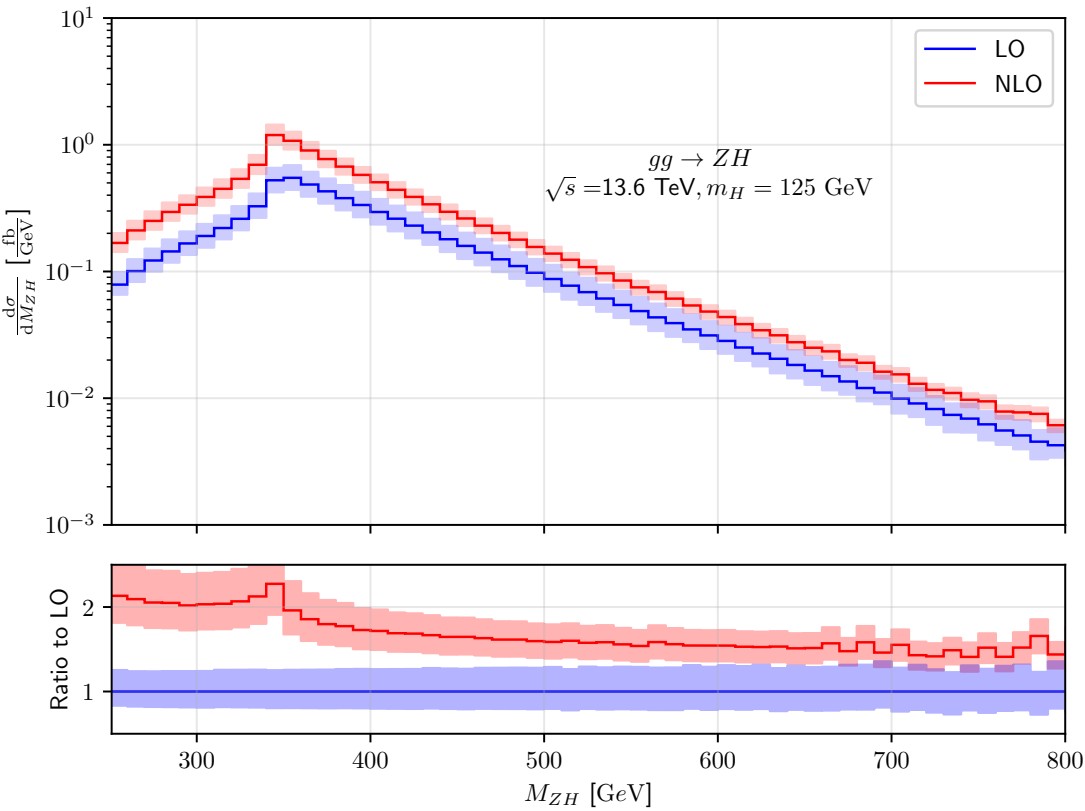

Figure 3: The invariant mass distribution of the $Z$-Higgs-system for an energy of $\sqrt{S} = 13.6$ TeV and a Higgs mass of $m_H = 125.0$ GeV. Presented are the NLO and LO distributions with the 7-points scale uncertainty.

We show results for the invariant-mass distribution in Fig. 3. We observe that the differential $K$-factor is not constant, with $K \sim 2$ for $M_{ZH} \lesssim 350$ GeV and then decreasing to roughly $K = 1.5$. Furthermore, the LO scale bands cannot account for the NLO corrections for any value of $M_{ZH}$ in the plot. We note that even considering a scale variation by a factor 3 in the LO does not give a scale uncertainty band that contains the NLO result [29].

### 3.3   Discussion of theoretical uncertainties

We observe that the main sources of theoretical uncertainties come from the QCD scale variation and from the choice of scheme and scale for the renormalisation of the top-quark mass. The QCD scale uncertainty is generally symmetric, and the inclusion of the NLO corrections leads to a reduction by roughly a factor of 2/3, with a residual error of about 15%. We recall that scale uncertainties of the YR4 results are at the 25% level[3] because they are based on the rescaling of a LO result: this is why they are comparable with those of $\sigma_{LO}$ in Tab. 1. The top-mass-scheme uncertainty is instead asymmetric, because the running of the top-quark mass in the $\overline{MS}$ scheme leads to a monotonous decrease of the value of $m_t$ used for the prediction, as discussed in Refs. [28, 33].

Compared to the ones discussed above, other typical sources of theoretical uncertainties are subdominant, and we quote a unique value for the LO and NLO results. The uncertainty associated to the value of $\alpha_s$ is estimated by varying $\alpha_s(m_Z)$ in Eq.(4) by $\pm 0.001$, and we

---

[3]see also Tab. 12 of Ref. [71]

find $\Delta_{\alpha_s} = \pm 0.7\%$ at LO and $\Delta_{\alpha_s} = \pm 1.8\%$ at NLO, where we averaged over all uncertainties for individual values of $m_H$ and $\sqrt{S}$. For the uncertainty due to the choice of the PDFs, we follow the prescription of Ref. [72], in particular we compute the cross-section for all provided Hessian PDF sets and compute

$$\delta^{\mathrm{PDF}}\sigma = \sqrt{\sum_{k=1}^{40}(\sigma^{(k)} - \sigma^{(0)})^2}, \tag{6}$$

where $\sigma^{(0)}$ is the cross-section for the standard PDF set. We find $\Delta_{\mathrm{PDF}} = \pm 1.3\%$ at LO and $\Delta_{\mathrm{PDF}} = \pm 2.3\%$ at NLO. Finally, we recall that the approach discussed in Sec. 2.1 for the calculation of the virtual corrections introduces an uncertainty within 1% due to the quality of our approximations.

## 4  Conclusion

The knowledge of higher-order corrections to $gg \to ZH$ is indispensable to decrease the impact of theoretical uncertainties on the hadronic cross section for associated $ZH$ production. In this report, we have produced improved theoretical predictions for $gg \to ZH$ including corrections at NLO in QCD. Despite the lack of an exact analytic calculation of the virtual corrections, the combination of complementary approximations allows to provide predictions that are reliable in the full phase space. In particular, we have used the combination of a forward expansion with a high-energy expansion, both supplemented with Padé approximants. The results of the exact numerical calculation have been essential to validate these approximations and to conclude that the error introduced by our analytic approach is below 1%: this is the smallest source of uncertainty in our predictions, at the same level as the $\alpha_s$ and PDF uncertainties.

When we compare our updated fixed-order predictions for $gg \to ZH$ with those used in the YR4 [21], the former are still dominated by scale uncertainties, which are around 15% at NLO. The central value of our results is also well compatible with the YR4 estimates at the inclusive level. The top-mass scheme uncertainty, which was not included in the YR4 predictions, also has a significant impact. The inclusion of even higher orders to reduce these theoretical uncertainties is therefore important. It would finally allow predictions for hadronic $ZH$ production with an accuracy of $\mathcal{O}(1\%)$, which is desirable for the High-Luminosity phase of the LHC.

## Acknowledgements

**Funding information**   We acknowledge support from the COMETA COST Action CA22130 and from the German Research Foundation (DFG) under grant 396021762 - TRR 257. R. G. was supported by the University of Padua under the 2023 STARS Grants@Unipd programme (Acronym and title of the project: HiggsPairs – Precise Theoretical Predictions for Higgs pair production at the LHC), the INFN Iniziativa Specifica AMPLITUDES and APINE, by the PNRR CN1- Spoke 2 and by the Italian Ministry of University and Research (MUR) Departments of Excellence grant 2023-2027 "Quantum Frontiers". P. P. G. is supported by the Ramón y Cajal grant RYC2022-038517-I funded by MCIN/AEI/10.13039/501100011033 and by FSE+, and by the Spanish Research Agency (Agencia Estatal de Investigación) through the grant IFT Centro de Excelencia Severo Ochoa No CEX2020-001007-S. L. C. was supported by the Natural Science Foundation of China under contract No. 12205171, No. 12235008, No. 12321005, and grants from Department of Science and Technology of Shandong province tsqn202312052

and 2024HWYQ-005. S. J. is supported by a Royal Society University Research Fellowship (URF/R1/201268) and the STFC under grant ST/X003167/1.

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
