# Peer review of "$gg \to ZH$ : updated predictions at NLO QCD"

_SciPost Physics Community Reports_

## Round 1 · Referee Report · Anonymous (Referee 1) · 2025-10-2

Strengths

1-This is a well written contribution, where the current status of the NLO corrections to $gg\to ZH$ is discussed.
2-The two independent computations of the NLO cross sections are carefully compared and new state of the art predictions are presented, which are based on the approximated analytic computation. The differences are shown to be negligible.
3-The discussion includes a detailed analysis of the remaining uncertainties.

Weaknesses

1-My only concern regards the interplay of the $gg\to ZH$ cross section with the Drell-Yan like production process. The two production mechanisms are not independent, they start to interfere at ${\cal O}(\alpha_S^2)$. The authors mention this in 2.2.1, when discussing the contributions driven by $Z$ boson radiation, but then provide their analysis for the $gg\to ZH$ cross section only, implicitly assuming that the two processes can be finally combined and treated as uncorrelated. I think this can only be an approximation, though presumably good. When scale variations are performed, shouldn't they (in principle) be carried out correlated with the Drell-Yan production mechanism ?

Report

I think the paper fully meets the acceptance criteria and should be published after the point I raised is addressed.

Requested changes

I think some additional comments on the interplay with the Drell-Yan production mechanism and the correlation of uncertainties would be appropriate.

Recommendation

Ask for minor revision

  • validity: high
  • significance: high
  • originality: good
  • clarity: high
  • formatting: excellent
  • grammar: excellent

Author:  Marco Vitti  on 2025-11-20  [id 6051]

(in reply to Report 1 on 2025-10-02)
Category:
answer to question

Dear Editor, dear referee

we thank the referee for carefully reading the manuscript and for the useful comments.

The referee requested a more prominent account of the interplay of the gluon-initated
channel with the Drell-Yan-like contribution. We addressed the request in the following
ways:

• We added a sentence at the end of the Introduction, in which we mention that
the two contributions cannot be formally treated as independent, and referring
to a discussion in Subsection 2.2.

• We added a paragraph at the beginning of Subsection 2.2, where we state (as
correctly pointed out by the referee) that real-emission diagrams that we included
in our O($\alpha_s^3$) corrections can interfere with Drell-Yan-like diagrams at O($\alpha_s^2$ ).

• We added a sentence at the end of Subsection 3.1, stating that our choice of
central scales for the evaluation of the gluon-initiated contribution does not take
into account correlations of scale uncertainties due to the interplay of the gluon-
initiated and Drell-Yan-like channels. However, we assume these effects to be of
higher order. We believe anyways that these effects would need further scrutiny
in the combination of the different channels.

We have submitted a revised version to the arXiv and we hope that, with the above
changes, it can be accepted for publication

Best regards,
Marco Vitti, for the authors

---

## Round 2 · Referee Report · Anonymous (Referee 1) · 2025-11-28

Strengths

1-This is a well written contribution, where the current status of the NLO corrections to gg→ZH is discussed.
2-The two independent computations of the NLO cross sections are carefully compared and new state of the art predictions are presented, which are based on the approximated analytic computation. The differences are shown to be negligible.
3-The discussion includes a detailed analysis of the remaining uncertainties.

Weaknesses

None

Report

I think the paper fully meets the acceptance criteria and can be now published

Requested changes

The authors have addressed the points raised in my first report.

Recommendation

Publish (easily meets expectations and criteria for this Journal; among top 50%)

---

## Round 2 · Author Response

The referee requested a more prominent account of the interplay of the gluon-initated channel with the Drell-Yan-like contribution. We addressed the request as described in the list of changes below.

---

## Round 2 · List of Changes

• We added a sentence at the end of the Introduction, in which we mention that
the Drell-Yan-like and the gluon-initiated contributions cannot be formally treated as independent, and referring
to a discussion in Subsection 2.2.

• We added a paragraph at the beginning of Subsection 2.2, where we state (as
correctly pointed out by the referee) that real-emission diagrams that we included
in our O($\alpha_s^3$ ) corrections can interfere with Drell-Yan-like diagrams at O($\alpha_s^2$ ).

• We added a sentence at the end of Subsection 3.1, stating that our choice of
central scales for the evaluation of the gluon-initiated contribution does not take
into account correlations of scale uncertainties due to the interplay of the gluon-
initiated and Drell-Yan-like channels. However, we assume these effects to be of
higher order.

---

## Editorial Decision

accepted_in_target_journal